# Predicting the Pathway Involvement of Compounds Annotated in the Reactome Knowledgebase

**DOI:** 10.3390/metabo15030161

**Published:** 2025-03-01

**Authors:** Erik D. Huckvale, Hunter N. B. Moseley

**Affiliations:** 1Markey Cancer Center, University of Kentucky, Lexington, KY 40536, USA; edhu227@uky.edu; 2Superfund Research Center, University of Kentucky, Lexington, KY 40536, USA; 3Department of Toxicology and Cancer Biology, University of Kentucky, Lexington, KY 40536, USA; 4Department of Molecular and Cellular Biochemistry, University of Kentucky, Lexington, KY 40536, USA; 5Institute for Biomedical Informatics, University of Kentucky, Lexington, KY 40536, USA

**Keywords:** reactome, metabolite, pathway, machine learning, supervised learning, binary classification, multilayer perceptron, biochemistry

## Abstract

**Background/Objectives:** Pathway annotations of non-macromolecular (relatively small) biomolecules facilitate biological and biomedical interpretation of metabolomics datasets. However, low pathway annotation levels of detected biomolecules hinder this type of interpretation. Thus, predicting the pathway involvement of detected but unannotated biomolecules has a high potential to improve metabolomics data analysis and omics integration. Past publications have only made use of the Kyoto Encyclopedia of Genes and Genomes-derived datasets to develop machine learning models to predict pathway involvement. However, to our knowledge, the Reactome knowledgebase has not been utilized to develop these types of predictive models. **Methods**: We created a dataset ready for machine learning using chemical representations of all pathway-annotated compounds available from the Reactome knowledgebase. Next, we trained and evaluated a multilayer perceptron binary classifier using combined metabolite-pathway paired feature vectors engineered from this new dataset. **Results**: While models trained on a prior corresponding KEGG dataset with 502 pathways scored a mean Matthew’s correlation coefficient (MCC) of 0.847 and a 0.0098 standard deviation, the models trained on the Reactome dataset with 3985 pathways demonstrated improved performance with a mean MCC of 0.916, but with a higher standard deviation of 0.0149. **Conclusions**: These results indicate that the pathways in Reactome can also be effectively predicted, greatly increasing the number of human-defined pathways available for prediction.

## 1. Introduction

Within cells and organisms, a wide range of small biomolecules (metabolites) are directly or indirectly associated with all biological processes that enable and sustain life. These associations represent a variety of molecular functions including biochemical reactions, but also various types of non-covalent molecular interactions. When the products of one reaction act as the reactants of another, they form paths of biochemical reactions within metabolic networks. Likewise, many non-covalent molecular interactions form paths in large molecular interaction networks that subsume the metabolic networks. These paths are described as human-defined biochemical and related pathways [1,2,3]. These pathways can be represented as graphs with chemical compounds (metabolite or xenobiotic) as nodes and reactions and non-covalent molecular interactions as edges. Technically, these reaction/interaction edges are often hyperedges within a hypergraph, but such hypergraphs are typically reduced to simpler graph representations. If a compound takes part in one of these reactions or interactions that are part of a pathway network, that compound is, by definition, directly associated with that pathway. Knowing the pathway associations of a compound is highly useful for biological and biomedical research both via pathway annotation enrichment analysis and visualization of molecular interaction networks that can provide insight into biological and disease mechanisms.

Knowledgebases such as the Kyoto Encyclopedia of Genes and Genomes (KEGG) [4] and Reactome [5] contain compound entries along with pathway annotations indicating the pathways that the compounds are associated with. However, most detectable compounds, even if they are in such knowledgebases, lack these pathway annotations. Due to these limitations, past work trained machine learning models to predict the pathway involvement of compounds using the known compound-to-pathway mappings along with compound molecular structures that are currently available. KEGG data, in particular, has been used in several past publications for training models to predict the pathway involvement of compounds based on their molecular structure [6,7,8,9,10]. Initially, these models only accepted features representing compounds, necessitating training a separate model for every pathway of interest and limiting the number of pathways that could feasibly be predicted. The limitations of this approach were overcome by the method introduced by Huckvale and Moseley, which trained a single binary classifier, accepting both compound and pathway features and predicting whether the given compound was associated with the given pathway [8]. This enabled a single model to predict an arbitrary number of pathways where the dataset is the concatenation outer product of the vector of compound features by the vector of pathway features rather than just the vector of compound features.

This combined metabolite-pathway feature vector approach, along with the significant increase in dataset size, enabled a state-of-the-art dataset and model to be constructed and trained on KEGG data, which contained 502 pathways and 6485 compound entries [10]. No longer being restricted to the number and type of pathways, this KEGG dataset was constructed from all the pathways with associated compounds annotated and was the largest dataset for this machine learning task to date, containing over three million entries. However, to our knowledge, the Reactome knowledgebase has not been used to construct a dataset for this pathway annotation prediction problem. Moreover, Reactome has thousands more annotated pathways than KEGG. Given the success with the prior KEGG dataset, we constructed an even larger dataset using all the pathways with compound association annotations available in Reactome. In this work, we demonstrate that a model can effectively predict the pathway involvement of compounds using Reactome data, greatly expanding the number of pathways that can be predicted.

## 2. Materials and Methods

The Reactome knowledgebase [5] provides pathway annotations, mapping pathway entry IDs to compound entry IDs where the compound entries are stored in a separate knowledgebase known as the Chemical Entities of Biological Interest (ChEBI) [11]. On 8 August 2024, we downloaded the molfiles [12] from ChEBI and the pathway annotations from Reactome in order to construct features and labels, respectively. We used the same dataset construction method as the state-of-the-art KEGG dataset [10], which was initially introduced by Huckvale and Moseley [8]. This method involved constructing feature vectors representing compounds from the information available in molfiles [12].

Using the atom coloring technique introduced by our lab [13,14,15] each feature corresponds to a molecular substructure, and the feature values are the number of occurrences of that substructure in a given compound. To construct the pathway feature vectors, the atom color counts are aggregated across all compounds associated with a given pathway. Both duplicate compound feature vectors and duplicate pathway feature vectors are removed, and then both the compound features and pathway features are normalized. Finally, the compound entries are cross-joined with the pathway entries to construct a dataset consisting of entries with combined compound and pathway features. Using the Reactome pathway annotations, we assigned to each entry a boolean label indicating whether the given compound is associated with the given pathway.

Table 1 compares attributes of the novel Reactome dataset to the prior KEGG dataset. We see that the Reactome dataset had less than a third of the number of compound entries (with pathway annotations) than the previous KEGG dataset. However, it had nearly eight times the number of pathway entries, resulting in more than double the total number of (cross-joined) entries. We also see that both the compound and pathway entries had a significantly lower number of features than the KEGG dataset, most likely because the Reactome dataset has far fewer compound entries. From past observations, the number of atom colors increases with a larger number and variety of compounds used [7,8,9,10].

Similar to KEGG, the Reactome pathways are organized in a hierarchy [16]. The pathways at the first level in the hierarchy, which we will call L1, are followed by level 2 (L2) pathways up to 9 pathway hierarchy levels. We will refer to these hierarchical levels as L1, L2, L3, L4, L5, and L6+, where L6+ refers to pathway levels L6, L7, L8, and L9 combined. Figure 1 shows the number of pathways within each hierarchical level. Appendix A shows this same information but includes L6, L7, L8, and L9 individually, where L9 only had 1 pathway within it, while L8 had 6, L7 had 24, and L6 had 184. Due to the small number of pathways within L9, L8, and L7, it was practical to combine them with L6 to form hierarchical level L6+, which contains enough pathways for statistically meaningful evaluations.

To compare the performance of different hierarchical levels, we additionally constructed two subsets of the Reactome dataset. We will refer to these subsets as L2+ and L3+. The L2+ dataset excludes the L1 pathways and includes the L2 pathways and all pathways under them in the hierarchy. Likewise, the L3+ dataset excludes the L1 and L2 pathways. The L1+ dataset includes all hierarchy levels and is therefore equivalent to the full Reactome dataset detailed in Table 1. Table 2 shows the number of pathway entries and the total number of entries in all three datasets.

We performed a cross-validation (CV) analysis on the full (L1+) Reactome dataset using 200 iterations that have some of the characteristics of both a bootstrap analysis and a jackknife analysis. The CV iterations each took 441 s (7.35 min) on average. For each CV iteration, we divided the entries into stratified train-test splits such that the proportion of positive entries in the test set is as close as possible to the proportion of positive entries in the train set on every CV iteration [17]. Next, we trained a multi-layer perceptron (MLP) binary classifier on the training set and evaluated it on the test set. The train/test set ratio was 9:1, equivalent to a ten-fold CV, but where only one-fold is tested. We collected the number of true positives (TP), true negatives (TN), false positives (FP), and false negatives (FN) for each compound in the test set, for each pathway in the test set, and for the entire test set. This enabled us to calculate metrics such as the Matthew’s correlation coefficient (MCC) for the entire test set in each CV iteration and a mean, median, and standard deviation across all CV iterations. This additionally enabled us to calculate an MCC for each individual compound and pathway by summing the TP, TN, FP, and FN across all CV iterations, constructing a single confusion matrix, and calculating a single overall metric value for each compound and pathway. While the entire test set was large enough to calculate a metric for each CV iteration, enabling the calculation of a mean value, this was not possible for individual pathways and compounds since a valid pathway or compound MCC cannot always be calculated for each CV iteration. In other words, a given train test split might not have sufficient positive entries for a single compound or pathway, resulting in a division by 0 when calculating MCC. Likewise, we performed the CV analysis on the L2+ and L3+ datasets, both using 50 iterations. This was a pragmatic decision, given the amount of computational resources required for these analyses. More CV iterations were performed on the L1+ dataset to ensure valid overall MCC (oMCC) scores for individual pathways and compounds, i.e., to avoid division by zero due to overall TP = 0.

The hardware used for this work included machines with up to 2 terabytes (TB) of random-access memory (RAM) and central processing units (CPUs) of 3.8 gigahertz (GHz) of processing speed. The name of the CPU chip was ‘Intel (R) Xeon (R) Platinum 8480CL’. The graphic processing units (GPUs) used had 81.56 gigabytes (GB) of GPU RAM, with the name of the GPU card being ‘NVIDIA H100 80 GB HBM3′.

All code for this work was written in major version 3 of the Python programming language [18]. Data processing and storage were conducted using the Pandas [19], NumPy [20], and H5Py [21] packages. Models were constructed and trained using the PyTorch Lightning [22] package built upon the PyTorch [23] package. The stratified train test splits were computed using the Sci-Kit Learn [24] package. Results were initially stored in an SQL database [25] using the DuckDB [26] package. Results were processed and visualized using Jupyter Notebooks [27], the Seaborn package [28] built upon the MatPlotLib [29] package, and the Tableau business intelligence application [30]. Model training and testing were profiled for GPU and CPU utilization using the gpu_tracker package [31].

## 3. Results

### 3.1. Main Results

Table 3 displays the mean, median, and standard deviation of the MCC across all of the CV iterations (200 iterations for L1+ and 50 iterations for L2+ and L3+). The MCC for each CV iteration was calculated from all the predictions in each test set. Specific hierarchical levels are filtered from the dataset based on the evaluation being performed. The L1+ dataset is the full dataset containing all hierarchical levels. The L2+ dataset excludes the L1 pathways and only contains the L2 pathways and onward. Likewise, the L3+ dataset contains L3 pathways and onward. We see a drop in overall performance as hierarchy levels are excluded. Appendix A includes these scores for all other metrics including accuracy, F1 score, precision, recall, and specificity.

Figure 2 shows the distributions of MCC of all test set predictions of 200 CV iterations for the full data set (L1+). The distribution appears unimodal, but left-skewed, explaining the slightly higher median MCC over the mean MCC.

Figure 3 shows the oMCC of the pathways in each hierarchical level by each dataset. The oMCC for each hierarchy level was calculated from a single confusion matrix containing the summed TP, TN, FP, and FN across all pathways within the given hierarchy level. The L1+ dataset results show the oMCC for all hierarchical levels, while the L2+ dataset did not contain the L1 pathways, so the confusion matrix for L1 pathways is not represented in the L2+ results. Likewise, the confusion matrix for neither L1 nor L2 pathways is represented in the L3+ dataset results. We see a decrease in MCC while getting deeper into the hierarchy. Appendix A shows this same information while including the confusion matrix counts (TP, TN, FP, and FN) for each hierarchy level from which the oMCC was calculated, as well as showing L6, L7, L8, and L9 separately.

Figure 4 provides the same information as Figure 3 but is arranged to compare differences between the datasets for each hierarchy level rather than differences between the hierarchy levels within each dataset. We see that for the L2 pathways, the L1+ dataset slightly outperformed the L2+ dataset, and their oMCC is exactly equal to the L3 pathways. For the L4, L5, and L6+ pathways, the L2+ dataset oMCC exceeds the L1+ datasets by no more than 0.005. The L3+ dataset consistently scores a lower oMCC than both the L1+ and L2+ datasets.

### 3.2. oMCC and Compound/Pathway Size

We define compound size as the number of non-hydrogen atoms within it. The size of a pathway is defined as the sum of the size of all the compounds associated with that pathway. Figure 5 shows the distribution of the size of the compound and pathway entries in the dataset.

The oMCC of an individual compound or pathway was calculated by summing the TP, TN, FP, and FN of that entry in the test sets across all 200 CV iterations of the L1+ dataset. Figure 6 shows the distribution of compound and pathway oMCC.

Figure 7 shows the distribution of the size of each pathway in each hierarchy level. We observe an overall trend of size decreasing deeper in the hierarchy. Note that the *y*-axis is on the log scale.

Figure 8 shows scatterplots comparing entry size to oMCC. We see that the maximum oMCC for compounds increases as compound size increases and does not reach 1.0 until a compound size of six non-hydrogen atoms. For pathways, we see that both maximum and minimum oMCC increase as pathway size increases. Additionally, we observe a funnel shape such that the variance of oMCC decreases as pathway size increases.

## 4. Discussion

While extensive work has been conducted for the machine learning task of predicting pathway involvement of compounds using KEGG data, this work demonstrates that the pathway annotations available in Reactome are also sufficient for this task, even more so. Table 4 compares the MCC of the Reactome dataset (Table 3) to that of the current state-of-the-art KEGG dataset [10]. We see that the Reactome pathways predict better than the KEGG pathways by over an 8% improvement in mean MCC. This level of improvement has a Cohen’s d effect size of over 4.6 and is clearly not due to random chance. While there are fewer compounds with pathway annotations in Reactome, there are many more pathways, resulting in an overall larger cross-joined dataset that is over twice the size of the KEGG dataset. The increased dataset size might explain the increased performance, though the nature of the pathway definitions may also be a factor. Figure 9 shows that KEGG pathways are more likely to be larger than Reactome pathways but also have more variance in pathway size than Reactome pathways. This higher pathway size variance is due to the bimodal distribution of pathway size seen in KEGG.

Using violin plots, Figure 10 better illustrates the distribution of MCC across 200 models trained on the KEGG and Reaction datasets, where each model corresponds to a single CV iteration. Models trained on the Reaction dataset clearly have better performance, but the distribution has slight bimodality, likely leading to the higher standard deviation as compared to the very unimodal KEGG model performance distribution. Both distributions have lower trailing tails.

As shown in Table 3, overall prediction performance improves with the inclusion of the higher-level pathways (i.e., L1 and L2). As shown in Figure 3, pathway-level prediction performance decreases with pathway depth. As shown in Figure 3 and Figure 4, the L3+ dataset consistently produced lower oMCC than the L1+ and L2+ datasets. This is consistent with past analyses on the KEGG dataset [10]. However, the relationship between the L1+ and L2+ datasets on oMCC is less obvious, as demonstrated in Figure 4. The L2 oMCC slightly improves with the L1+ dataset, but the L3 oMCC is equivalent for L1+ and L2+ datasets. However, the L4 and deeper oMCC are all slightly better with the L2+ dataset. The improvements are minor, with the L2+ dataset not increasing MCC by any more than 0.005. However, we think the use of the full dataset is warranted since the overall improvement in mean MCC of the L1+ dataset over the L2+ dataset is 0.009. But in the future, it may be advantageous to train separate models optimized for different ranges of hierarchical levels.

As shown in Figure 7, we see that pathway size decreases overall as the hierarchical level increases, i.e., deeper into the pathway hierarchy defined by Reactome. As shown in Figure 8, we also observe a relationship between pathway size and oMCC where smaller pathways are more difficult to predict than larger pathways (e.g., we do not see a maximum pathway oMCC of 1.0 until reaching a pathway size of 39). This explains why we see a decrease in oMCC for pathways deeper in the hierarchy. We also observe greater variance of oMCC in smaller pathways, indicating increased robustness of larger pathways. We see similar trends between compound size and compound oMCC (e.g., a maximum oMCC not reaching 1.0 until reaching a compound size of 6). These results are consistent with past analyses of the KEGG dataset [10]. Users should keep these trends in mind when predicting smaller compound and pathway combinations.

## Figures and Tables

**Figure 1 metabolites-15-00161-f001:**
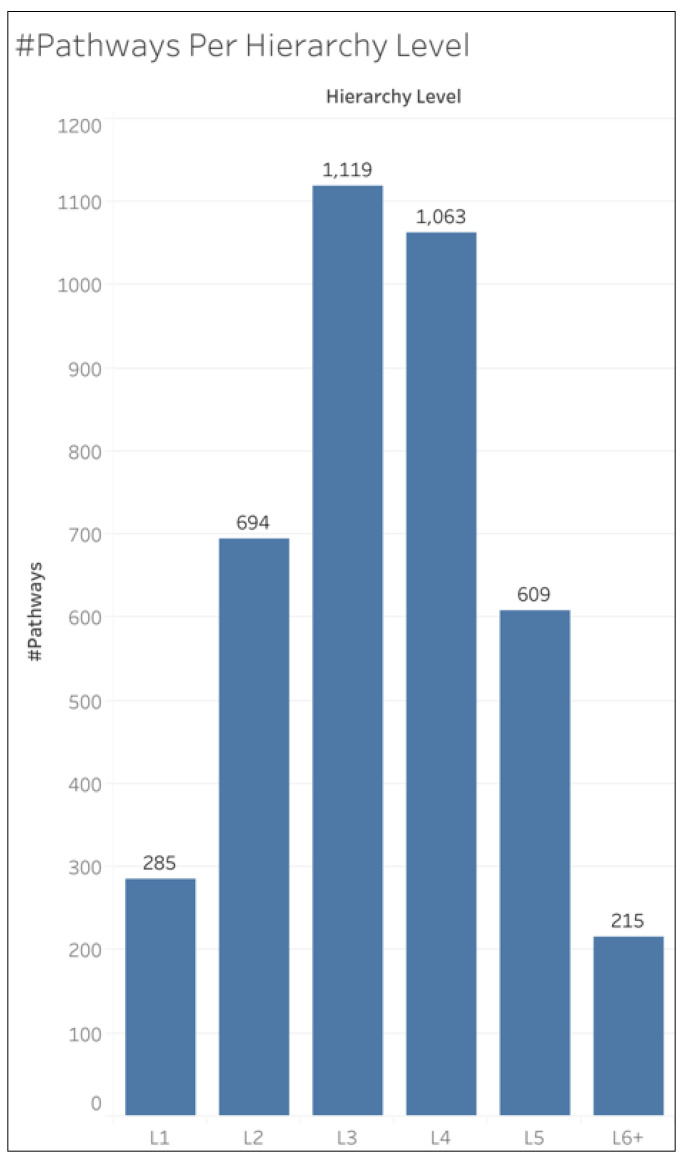
Number of pathways per hierarchy level. Reactome organizes its pathways into a hierarchical structure beginning at the L1 pathways, which have L2 pathways under them, which have L3 under them, etc. L6+ refers to hierarchy levels L6, L7, L8, and L9 combined.

**Figure 2 metabolites-15-00161-f002:**
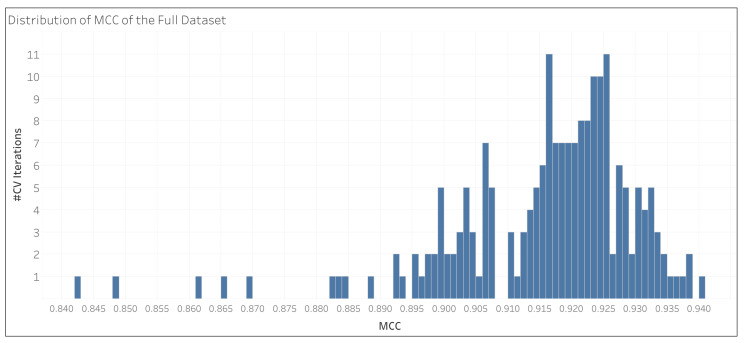
Distribution of MCC of the full dataset. The full dataset contained all pathway hierarchy levels. The distribution of MCC for 200 CV iterations is displayed.

**Figure 3 metabolites-15-00161-f003:**
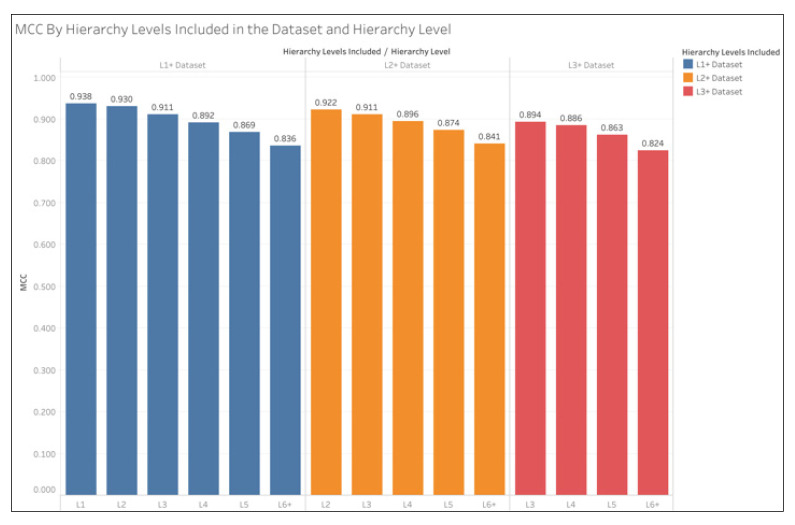
oMCC by dataset and hierarchy level. The L1+ dataset is the full dataset, while L2+ excludes the L1 pathways, and L3+ excludes the L1 and L2 pathways. MCC is calculated from the sum of TP, TN. FP, and FN across all pathways and CV iterations in each hierarchy level.

**Figure 4 metabolites-15-00161-f004:**
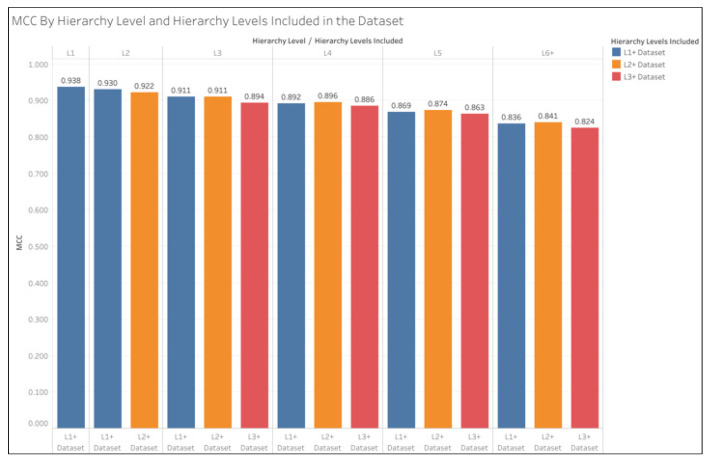
oMCC by hierarchy level and dataset. The L1+ dataset is the full dataset, while L2+ excludes the L1 pathways, and L3+ excludes the L1 and L2 pathways. oMCC is calculated from the sum of TP, TN. FP, and FN across all pathways and CV iterations in each hierarchy level.

**Figure 5 metabolites-15-00161-f005:**
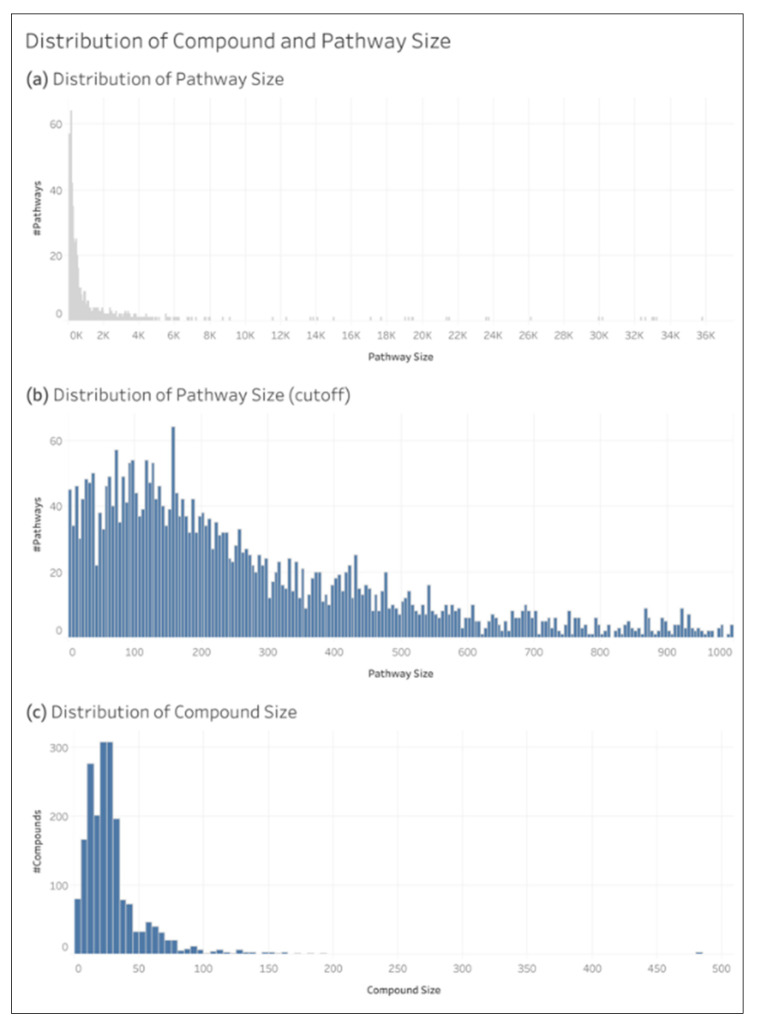
Distributions of compound size and pathway size.

**Figure 6 metabolites-15-00161-f006:**
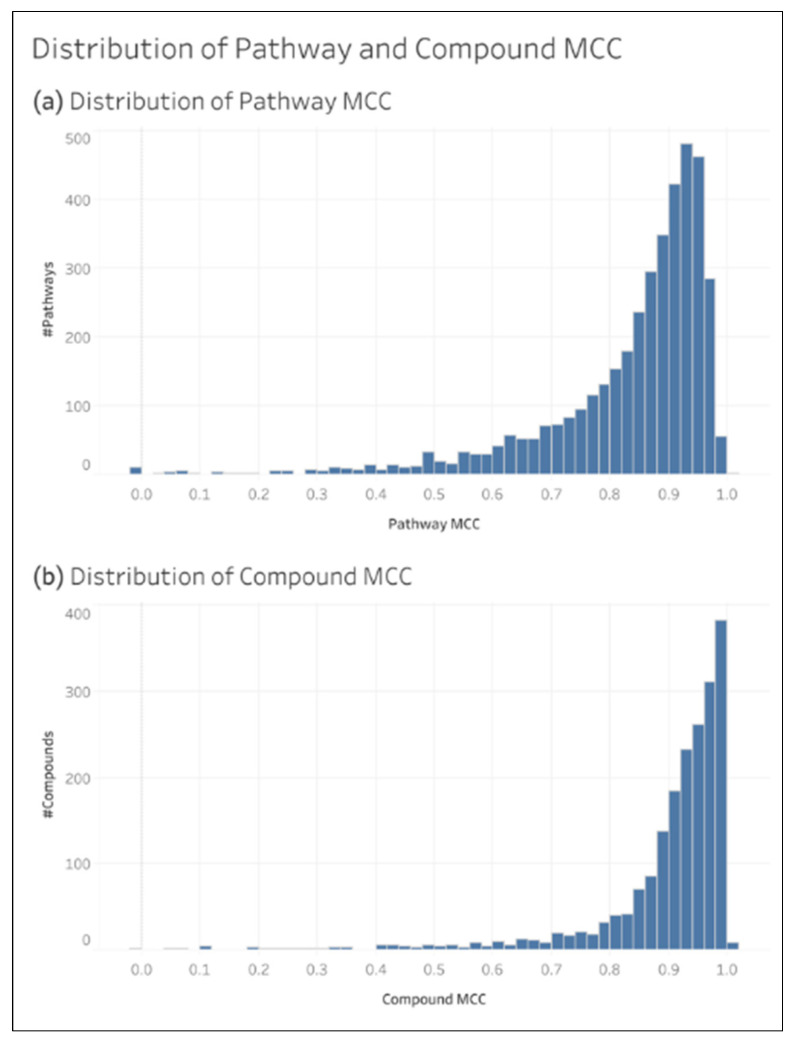
Distributions of individual compound oMCC and individual pathway oMCC.

**Figure 7 metabolites-15-00161-f007:**
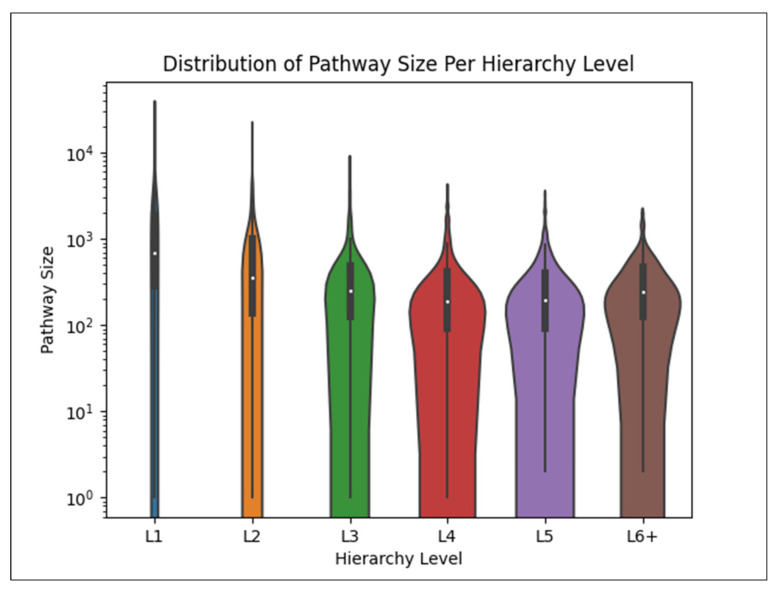
Violin plot displaying the distribution of the sizes of pathways in each hierarchy level.

**Figure 8 metabolites-15-00161-f008:**
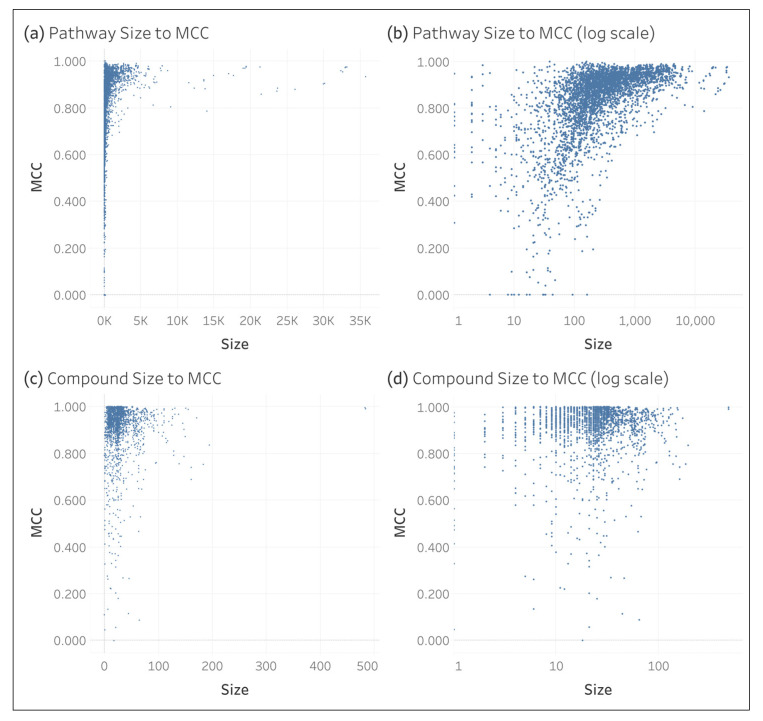
Individual compound and pathway compared to oMCC.

**Figure 9 metabolites-15-00161-f009:**
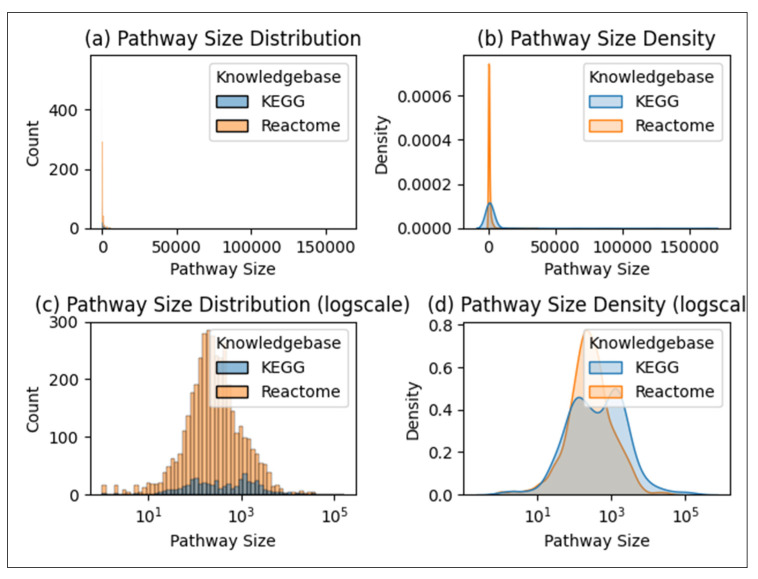
The distribution and probability density of pathway size among KEGG and Reactome pathways.

**Figure 10 metabolites-15-00161-f010:**
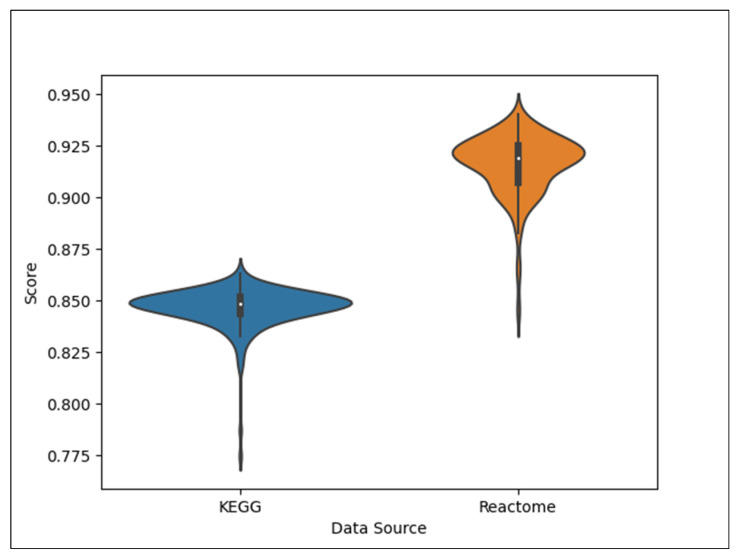
Distribution of MCCs for models trained on the KEGG and Reactome datasets. Notice the *y*-axis range between 0.75 and 0.975.

**Table 1 metabolites-15-00161-t001:** Size of the novel Reactome dataset compared to the prior KEGG dataset.

Dataset	# Compounds	# Pathways	# Compound Features	# Pathway Features	# Entries
KEGG	6485	502	16,509	11,321	3,255,470
Reactome	1976	3985	6187	5386	**7,874,360**

The Reactome dataset consisted of compound entries and pathway entries, both of which are feature vectors representing compounds and pathways, respectively. The two were cross-joined into combined compound and pathway feature vectors, resulting in 1976 × 3985 = 7,874,360 total entries. The Reactome dataset has fewer compounds but more pathways than the prior KEGG dataset. Overall, the Reactome dataset has more than double the number of entries as the KEGG dataset.

**Table 2 metabolites-15-00161-t002:** Dataset sizes by the pathway hierarchy levels included.

Dataset	# Pathways	# Entries
L1+	3985	7,874,360
L2+	3700	7,311,200
L3+	3006	5,939,856

The L1+ dataset is the full Reactome dataset since it contains the pathways from the first hierarchy level (L1) and onwards (i.e., all hierarchy levels). L2+ excludes the L1 pathways, as it contains the L2 pathways and onwards. Likewise, L3+ excludes the L1 and L2 pathways.

**Table 3 metabolites-15-00161-t003:** MCC by hierarchy levels included in the dataset.

Hierarchy Levels Included	Mean MCC	Median MCC	Standard Deviation
L1+	0.916	0.919	0.0149
L2+	0.907	0.907	0.0099
L3+	0.884	0.886	0.0134

Displayed are the mean, median, and standard deviation MCC of all test set predictions across CV iterations. L1+ contains all pathway hierarchy levels, while L2+ contains L2 and beyond, and L3+ contains L3 and beyond.

**Table 4 metabolites-15-00161-t004:** Comparing the MCC of the prior KEGG dataset to the L1+ Reactome dataset.

Dataset	Mean MCC	Median MCC	Standard Deviation
Reactome	0.916	0.919	0.0149
KEGG	0.847	0.848	0.0098

## Data Availability

Code and data for fully reproducing the results in this work are available at https://doi.org/10.6084/m9.figshare.27478065.

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
