# Peer review of "Predicting the Pathway Involvement of Compounds Annotated in the Reactome Knowledgebase"

_metabolites, 2025, doi:10.3390/metabo15030161_

Round 1

Reviewer 1 Report

Comments and Suggestions for Authors

With great pleasure, I have reviewed another work by your experienced team. The paper is well-written, with a solid structure, appropriate selection of literature, logical reasoning, and sound conclusions, all of which raise no objections.

However, I have some comments regarding the opening sentences of the manuscript. The sentences:

"Within cells and organisms, a wide range of small biomolecules (metabolites) are directly or indirectly associated with all biological processes that enable and sustain life. These associations represent a variety of molecular functions, including biochemical reactions, but also various types of non-covalent molecular interactions." (line 35) – are confusing. I understand the authors' intention, and the meaning is clear. However, the phrasing resembles a textbook. I suggest rewording this part of introduction.

Additionally, the text has not been carefully proofread after revisions. There are strikethroughs in table descriptions (lines 111, 182, 242) and yellow highlights (line 273). I strongly recommend a thorough proofreading of the text.

One last minor remark: the reference (link) to the Reactome pathway database (line 113) should, in my opinion, be moved to the References section.

Author Response

Reviewer 1:

With great pleasure, I have reviewed another work by your experienced team. The paper is well-written, with a solid structure, appropriate selection of literature, logical reasoning, and sound conclusions, all of which raise no objections.

Response:

We thank the reviewer for their very positive comments. We have worked very hard to generate this manuscript and the results within it.

Issue 1:

However, I have some comments regarding the opening sentences of the manuscript. The sentences:

"Within cells and organisms, a wide range of small biomolecules (metabolites) are directly or indirectly associated with all biological processes that enable and sustain life. These associations represent a variety of molecular functions, including biochemical reactions, but also various types of non-covalent molecular interactions." (line 35) – are confusing. I understand the authors' intention, and the meaning is clear. However, the phrasing resembles a textbook. I suggest rewording this part of introduction.

Response:

We acknowledge that this text is a bit dry and states the obvious. However, we wanted a concise description of metabolites and their role in living processes for those less familiar with both these concepts. We are also trying to bridge from familiar metabolic concepts into more computational concepts related to graphs and networks. Respectfully, we are not sure how to accomplish both any better than we have. If the reviewer has specific suggestions, we will try to incorporate them.

Issue 2:

Additionally, the text has not been carefully proofread after revisions. There are strikethroughs in table descriptions (lines 111, 182, 242) and yellow highlights (line 273). I strongly recommend a thorough proofreading of the text.

Response:

We have removed the yellow highlight; however, we did not see strike throughs in the table descriptions nor anywhere else. We also checked the version of the manuscript we submitted and did not see any strike throughs. Those may have been inadvertently added by the journal in a version of the manuscript provided to you.

Issue 3:

One last minor remark: the reference (link) to the Reactome pathway database (line 113) should, in my opinion, be moved to the References section.

Response:

Thanks for the suggestion. We have done this.

Reviewer 2 Report

Comments and Suggestions for Authors

The authors described the prediction of the pathway involvement of the compounds annotated in the Reactome knowledge base. The presentation of the manuscript is good. My comments are

  1. What was the average training time per model iteration for the hardware specified in the paper, and how scalable is the approach for larger datasets?
  2. Is there any specific reason to choose Matthew’s correlation coefficient as the primary evaluation metric?
  3. Authors can explore model explainability techniques like SHAP or LIME to interpret individual pathway predictions.
  4. How does the hierarchical structure of Reactome pathways influence model accuracy at different levels?
  5. Is the feature importance analysis technique employed useful for finding any molecular substructures that were strongly associated with pathway involvement?
  6. Did the authors try any other loss function apart from binary cross-entropy?
  7. Can model compression techniques or knowledge distillation be employed to reduce computational load while maintaining accuracy?

Author Response

Reviewer 2:

The authors described the prediction of the pathway involvement of the compounds annotated in the Reactome knowledge base. The presentation of the manuscript is good. My comments are

Response:

We thank the reviewer for their comments.

Issue 1:

1. What was the average training time per model iteration for the hardware specified in the paper, and how scalable is the approach for larger datasets?

Response:

We have added this detail:

“The CV iterations each took 441 seconds (7.35 minutes) on average”.

However, the training time is very dependent on both the model and the training set size. But the approach presented here does appear to be very scalable.

Issue 2:

2. Is there any specific reason to choose Matthew’s correlation coefficient as the primary evaluation metric?

Response:

It better represents overall model performance since it includes all aspects of the confusion matrix, especially when datasets are unbalanced. The F1-score ignores true negatives, which is a problem when false positives and false negatives are very different. This is machine learning 101.

Issue 3:

3. Authors can explore model explainability techniques like SHAP or LIME to interpret individual pathway predictions.

Response:

There are a lot of things we can explore. But we are most interested in the overall performance as we generate larger datasets. We actually explored pathway-specific feature importance in prior publications on smaller datasets. See our paper:

Erik D. Huckvale and Hunter N.B. Moseley. "Predicting the Pathway Involvement Of Metabolites Based on Combined Metabolite and Pathway Features" Metabolites 14, 266 (2024). https://doi.org/10.3390/metabo14050266

Issue 4:

4. How does the hierarchical structure of Reactome pathways influence model accuracy at different levels?

Response:

We show model performance per pathway hierarchy level in Table 3, Figure 3, and Figure 4. Table S2 shows the confusion matrix for each hierarchy level. Table S1 shows the accuracy for the L1+, L2+, and L3+ datasets. If anyone wants more detail than this, the full results are in the Figshare item: https://doi.org/10.6084/m9.figshare.27478065

Issue 5:

5. Is the feature importance analysis technique employed useful for finding any molecular substructures that were strongly associated with pathway involvement?

Response:

We have not done any feature importance analysis in this work. Neural networks are notoriously difficult to identify important features. For many years, it is been considered practically impossible to do. But there is a recent paper on generating feature importance from MLPs: https://dl.acm.org/doi/10.1007/s10115-023-01959-7
However, this requires changing the model, which may perturb hyperparameter tuning and is clearly beyond the scope of this manuscript. As mentioned in Issue 3, we have previously explored feature importance in related datasets small enough for us to effectively use XGBoost.

Issue 6:

6. Did the authors try any other loss function apart from binary cross-entropy?

Response:

We have explored additional loss functions in prior work and found binary cross-entropy to be the best with this classification problem.

Issue 7:

7. Can model compression techniques or knowledge distillation be employed to reduce computational load while maintaining accuracy?

Response:

We are exploring graph attention networks; however, there is a fundamental problem of fitting both the model and the dataset within available GPU RAM, which is absolutely needed for efficient training. We are doing this work on NVIDIA H100s with 80GB RAM, so it is not an issue of access to high-end GPUs. Please look at the gpu_tracker software we have developed for profiling GPUs in an HPC environment.

Erik D. Huckvale and Hunter N.B. Moseley. "gpu_tracker: Python package for tracking and profiling GPU utilization in both desktop and high-performance computing environments" arXiv arXiv:2404.01473 (2024). https://arxiv.org/abs/2404.01473 https://github.com/MoseleyBioinformaticsLab/gpu_tracker

We developed this package explicitly to optimize the models presented in this manuscript.

Reviewer 3 Report

Comments and Suggestions for Authors

As you can see, Reactome's various annotations and KEGG data identify better compounds for the improvement pathway. This collection of machine learning, medical, and scientific models is fantastic. I was astounded by the prediction models and datasets when I was investigating them. They contain an open-source database of scientific debates and case studies on human biological processes, including reactions and pathways. It is requested that you include additional machine learning or deep learning mechanisms in a future study or paper so that you can see future variations of the data that is currently available within Gene Ontology controlled vocabularies to describe biological processes and molecular functions. To increase prediction rates and accuracy, you might be able to use some quantum machine learning models with your datasets. These cutting-edge methods may improve the capacity to identify intricate relationships and patterns in the data, which could result in groundbreaking findings in personalized medicine and genomics. By combining these cutting-edge methods, scientists might find fresh perspectives that influence the development of biological knowledge and treatment plans in the future. Overall, it is a great paper

Author Response

Reviewer 3:

As you can see, Reactome's various annotations and KEGG data identify better compounds for the improvement pathway. This collection of machine learning, medical, and scientific models is fantastic. I was astounded by the prediction models and datasets when I was investigating them. They contain an open-source database of scientific debates and case studies on human biological processes, including reactions and pathways.

Response:

We thank the reviewer for their very positive comments! Yes, we agree, Reactome is an excellent pathway knowledgebase for annotation enrichment analysis! We use it all the time.

Issue 1:

It is requested that you include additional machine learning or deep learning mechanisms in a future study or paper so that you can see future variations of the data that is currently available within Gene Ontology controlled vocabularies to describe biological processes and molecular functions.

Response:

We do plan to apply our pathway prediction models to demonstrate improvement in pathway enrichment analyses in future publications. This is the driving goal for the current manuscript.

The problem with using Gene Ontology is that annotations are specific to gene(-products) and not to metabolites. You can try to associate a metabolite with an enzyme and then to a GO term, but KEGG, Reactome, and MetaCyc all provide pathway annotations to specific metabolites.

Issue 2:

To increase prediction rates and accuracy, you might be able to use some quantum machine learning models with your datasets. These cutting-edge methods may improve the capacity to identify intricate relationships and patterns in the data, which could result in groundbreaking findings in personalized medicine and genomics. By combining these cutting-edge methods, scientists might find fresh perspectives that influence the development of biological knowledge and treatment plans in the future. Overall, it is a great paper

Response:

We do not have access to the necessary hardware (quantum computers) to train and test quantum machine learning models. But we are exploring cutting edge deep learning methods like graph attention networks (GAT). The problem is fitting a GAT designed for a double feature vector along with the dataset within GPU RAM, so we can efficiently train the model on the NVIDIA H100s that we have access to. Without the 20-fold efficiency gains, it would take over a year to train such models with the necessary hyperparameter tuning, given our hardware constraints.

Again, thank you very much for the very positive comments!